# Redox Signaling in Endosomes Using the Example of EGF Receptors: A Graphical Review

**DOI:** 10.3390/antiox13101215

**Published:** 2024-10-09

**Authors:** Dana Maureen Hebchen, Katrin Schröder

**Affiliations:** Institute of Physiology, Medical Faculty, Goethe University, Theodor-Stern-Kai 7, 60590 Frankfurt am Main, Germany; hebchen@vrc.uni-frankfurt.de

**Keywords:** early endosomes, redoxosomes, *trans*-activation, EGFR, reactive oxygen species

## Abstract

Early endosomes represent first-line sorting compartments or even organelles for internalized molecules. They enable the transport of molecules or ligands to other compartments of the cell, such as lysosomes, for degradation or recycle them back to the membrane by various mechanisms. Moreover, early endosomes function as signaling and scaffolding platforms to initiate or prolong distinct signaling pathways. Accordingly, early endosomes have to be recognized as either part of a degradation or recycling pathway. The physical proximity of many ligand-binding receptors with other membrane-bound proteins or complexes such as NADPH oxidases may result in an interaction of second messengers, like reactive oxygen species (ROS) and early endosomes, that promote the correct recognition of individual early endosomes. In fact, redoxosomes comprise an endosomal subsection of signaling endosomes. One example of such potential interaction is epidermal growth factor receptor (EGFR) signaling. Here we summarize recent findings on EGFR signaling as a well-studied example for receptor trafficking and *trans*-activation and illustrate the interplay between cellular and endosomal ROS.

## 1. Introduction

The plasma membrane of a cell is a highly dynamic structure with a permanent constitutive turnover, mainly facilitated by endosomes. For example, macrophages and fibroblasts internalize more than 200% of their entire surface area per hour [1]. A major task of the cellular membrane is to keep contact with the surrounding cells and tissue. In order to maintain environmental contact, cells express receptors at their surface to measure mechanical distortions or pressure to sense circumstances such as pH or temperature and eventually obtain information or instructions in a ligand-mediated way. Those ligands usually find a specialized receptor, and binding of both results in specific signaling pathways. Receptor-mediated signaling is classified as listed in Table 1.

Excellent reviews have been published on type I and II signaling [2,3]. In this review, we will concentrate on type III signaling on the example of epithelial growth factor (EGF) and its receptor (epithelial growth factor receptor, EGFR), which involves receptor internalization.

Early endosomes are part of the endocytic trafficking pathways for cargo molecules internalized from the plasma membrane [4]. Internalization is carried out by diverse mechanisms, including clathrin-dependent endocytosis, phago- and pinocytosis, or routes utilizing dynamin, caveolae, and others [5]. In all cases, early endosomes serve as the primary sorting platform for diverse cargo, for example receptors and their ligands [6,7,8].

The intention of this graphical review is to provide an overview of early endosomes in particular of key signaling components. In particular, we will address the endosomal sorting and trafficking of EGFR. Eventually, we will delineate the concept of redox-active endosomes (redoxosomes) using the example of EGFR *trans*-activation by endosomal reactive oxygen species (ROS).

### 1.1. Models of the Early Endosome

For the description of endocytic traffic (Figure 1), two basic models have been evolved, namely the “pre-existing compartment model” in contrast to the “maturation model” which are both disputed [9,10]. Briefly, the “pre-existing compartment model” aka “Endosomal carrier vesicle hypothesis” states constant endosomal compartments, e.g., early endosomes and late endosomes with stable marker proteins. Consequently, cargo is shuttled through transport vesicles [9]. Conversely, in the “maturation model”, a single structure gradually transforms into a functionally different entity. In the latter case, regulatory proteins are rather recruited from the cytoplasm than resident at the endosome [6,11]. Both models agree on initial fusion events between primary endocytic vesicles and early endosomes followed by cargo sorting but differ in the origin of later endosomal structures and their identity over time [10].

### 1.2. Structure of the Early Endosome

Early endosomes are dynamic and morphologically complex structures (Figure 2) displaying a *cis-trans* polarity. On the *cis*-side, incoming vesicles are received, whereas the *trans*-side shows both tubular (~60 nm diameter, pH~6.5) and multivesicular (~300 nm diameter, pH~6.3–6.8) regions as functional subdomains [4]. Cargo molecules are sorted in the early endosomes either to degrading or recycling routes; however, in most cases the sorting signals are poorly understood [6].

Functional areas within early endosomes are likely organized by membrane-binding proteins like annexins interacting as well with the actin cytoskeleton and cholesterol-enriched rafts. Apart from shuttling vesicles budded off from the early endosomes, motor protein complexes, including myosins, kinesins, and dynein, control early endosomes’ intracellular movement [7,8].

In spite of the challenges in identifying unique molecular markers for each endosomal component, Ras-associated proteins (Rab) and their effectors are essential in regulating endocytic traffic. In the early endosomes, especially Rab4,5,10,14,21,22 are prominent [8]. In the following section, we will exemplarily consider some key effectors of Rab4 and Rab5, although an exhaustive analysis of the complete signaling network at the early endosomes is beyond the scope of this review.

Early endosomes are complex structures with distinct functional regions for receiving, sorting, and processing cargo molecules, organized by membrane-binding proteins and controlled by motor protein complexes for intracellular movement. Rab proteins and their effectors, particularly Rab4,5,10,14,21,22 in early endosomes, play crucial roles in regulating endocytic traffic, though unique molecular markers for each endosomal component remain challenging to identify.

### 1.3. Signaling Pathways at Early Endosomes

As mentioned above, Rab proteins are involved in all aspects of vesicle processing (budding, fusion, tethering, etc.) (Figure 2). In their active GTP-bound form, Rabs recruit specific effector proteins. Rab5-GTP promotes its own activation through a positive feedback circuit with Rab GTPase-binding effector protein 1 (Rabaptin-5) and Rab5 GDP/GTP exchange factor 5 (Rabex-5). Additionally, Rabaptin-5 plays a role in vesicle budding from the early endosomes by interacting with the clathrin adapter AP-1 [7].

Among the Rab5 effectors in early endosomes, early endosomal antigen-1 (EEA1) is commonly used as an early endosome marker [12]. EEA1 and Rab5 also participate in redoxosomes discussed in a later section of this review. In concert with soluble N-ethylmaleimide-sensitive factor attachment protein receptors (SNAREs), EEA1 mediates endosome fusions. Furthermore, EEA1 binds to phosphatidylinositol-3-phosphate (PtdIns3P), synthesized by the phosphatidylinositol 3-kinase Vps34 complex [7]. In general, spatially restricted lipid synthesis and enrichment allow the selective recruitment of PtdIns3P-binding proteins like the NADPH oxidase subunit p40^phox^ [4]. Similarly, PtdIns3P is converted to phosphatidylinositol-3,5-bisphosphate (PtdIns3,5P2) by the 1-phosphatidylinositol 3-phosphate 5-kinase (PIKFyve) complex. PIKFyve and its counterpart, the lipid phosphatase myotubularin (MTM1), tightly control PtdIns3P levels, which is crucial for proper EGFR sorting (see later section) [6]. Contrary to Rab5, Rab4 is not only localized on early endosomes but also on RE. In the first scenario, Rab4 and Eps15 homology domain (EHD) proteins coordinate the exit of cargo to the perinuclear endocytic recycling compartment (ERC), which is part of the “slow-recycling” pathway in contrast to the “short loop” transport directly back to the plasma membrane [8]. Rab4 and Rab5 are linked by dual effectors, e.g., Rab interacting protein 4 (RabIP4) and Rabenosyn-5 [4].

Although cargo sorting will not be covered in detail here, notably, the multiple subdomains inside the early endosomes permit cargo sorting to various organelles, inter alia the trans-Golgi network. In this context, the multimeric retrograde transport machinery is initiated at the early endosomes through the action of sorting nexins (SNX) with other components of the retromer [8].

Interestingly, Rab effectors may partake in signal transduction as well, as it was shown for adapter proteins containing the PH domain, PTB domain, and leucine zipper motif 1,2 (APPL1,2), which remodel nucleosomes and regulate the activity of the kinase Akt [7,13].

A growing body of evidence links early endosomes to signal transduction of internalized receptors. Internalization routes, ligand type, and concentration determine the signaling outcome. In addition, signaling after endocytosis can be different to the plasma membrane depending on the specific protein equipment of the endosome [12].

Rab proteins, particularly Rab5 and its effectors like EEA1, play crucial roles in early endosome functions, including vesicle processing, fusion, and cargo sorting, while also participating in complex signaling networks and lipid metabolism. Early endosomes are increasingly recognized as important platforms for signal transduction of internalized receptors, with the specific internalization routes, ligand types, and endosomal protein composition influencing signaling outcomes.

## 2. The Endosomal Compartment in EGF Signaling

Binding of a ligand to type III receptors triggers internalization and recycling of the receptor itself as well as surrounding portions of the membrane. Internalization of the receptor represents a mechanism to control the dynamic of EGF signaling, as internalization of the EGFR and therefore reduction of the number of receptors at the cell surface at least transiently reduces the ability of its ligand EGF to spur further signaling [14].

Internalization of EGFR is stimulated after ligand-binding and *trans*-phosphorylation of the receptor tyrosine kinases [15]. We will not describe the mechanisms of EGFR activation and the downstream signaling cascades in this review; however, it should be noted that some EGFR signaling is supposed to not only be continued in early endosomes, but also several pathways seem to rely on endocytosis [13,16].

Internalization requires fusion of membrane cargo vesicles to organelles called endosomes (Figure 3). Those can be classified as early, sorting, recycling, or late endosomes depending on their stage of maturation [17,18]. Eventually lysosomes are considered to be part of the endosomal compartment [19]. As such, early endosomes function as signaling and scaffolding platforms to initiate or prolong distinct signaling pathways, or degradation of the internalized ligand and receptor. Accordingly, recognition of early endosomes as part of a degradation, signaling, or recycling pathway represents a major step in ligand-induced signal transduction.

Endocytosis involves or even depends on the GTPase dynamin 2, which forms helical polymers at lipid rafts at the necks of budding vesicles, and its GTP hydrolysis-dependent conformational changes promote fission of the tubular membrane in order to generate a free endocytic vesicle [20]. According to the various nature of lipid rafts, two major models of endosome formation are proposed: clathrin-dependent and -independent endocytosis. Clathrin-dependent endocytosis most often is described to build vesicles from 100–200 nm, while clathrin-independent endocytosis is mediated by caveolins (caveolae) and lipid rafts and involves smaller vesicles (40–80 nm) [19]. In the case of EGFR (Figure 4(1)), endocytosis is primarily clathrin-dependent, and around 20–25% of clathrin-coated pits are loaded with EGF at 37 °C, no matter how high an EGF concentration is [21]. In fact, in different strains of human fibroblasts, each cell contains 40,000–100,000 binding sites for EGF [14]. The data indicates a saturated internalization of EGF determined by the number of EGF binding sites. Interestingly, EGF-loaded vesicles lose their clathrin coat as a prerequisite for further sorting and signaling. In general, internalized ligand-receptor complexes after endocytosis via clathrin-coated pits enter the endosomal compartment. EGFR endocytosis is subject to controversy concerning the conditions and proteins involved; some studies suggest that EGFR is internalized via clathrin-dependent and -independent pathways [16,22]. Furthermore, the dimerization partners influence the sensitivity to downregulation [23].

Once in early endosomes, membrane proteins are rapidly recycled to the plasma membrane, while fluid-phase proteins usually proceed towards the lysosome (Figure 4) [24].

A major peculiarity of early endosomes is their ability to facilitate recycling of non-ubiquitinated endocytosed receptors, while ubiquitinated receptors proceed to lysosomal degradation. Recycling endosomes are concentrated at the microtubule organizing center and consist of a largely tubular network [15]. While late endosomes are mainly spherical, lack tubules, and contain many closely packed intraluminal vesicles, early endosomes consist of a dynamic tubular-vesicular network with vesicles up to 1 µm in diameter and with connected tubules of approx. 50 nm diameter. For maturation of early endosomes into late endosomes, they become increasingly acidic (pH ≈ 6.0–6.8) maintained by an ATP-driven proton pump [25] or anion transporter chloride channels such as CLC3 [26], which results in the dissociation of the ligand from its receptor. Sorting of endosomes is mediated by Rab GTPases, with Rab5 mainly expressed in early endosomes. Other Rab family members, Rab4 and Rab11, facilitate the labeling of early endosomes as they generate recycling vesicles that fuse with the plasma membrane, returning mostly non-ubiquitinated receptors directly back to the cell surface. In contrast, Rab11-positive but Rab4-negative pericentriolar recycling endosomes appear to be important for membrane domain maintenance and membrane protein mobilization [27]. Within early endosomes, most receptors disengage from their ligands, with subsequent recycling of the receptor and degradation of the ligand. In the case of EGFR, disengagement is determined by the ligand bound. TGFα/EGFR is disengaged at endosomal pH, while EGF/EGFR is not. Different from LDL receptors, EGF and the EGFR have a pH-resistant bond that persists until it is delivered to lysosomes for their degradation [28,29]. In both cases, adapter proteins like growth factor receptor-bound protein 2 (Grb2) and SH3 domain-containing kinase-binding protein 1 (CIN85) associate with the activated EGFR, leading to recruitment of the E3 ubiquitin (ub) ligase Cbl [30]. Next, Ub-binding proteins, for example, the clathrin coat, tether to Ub- chains at the EGFR. In early endosomes, Cbl proceeds with EGFR ubiquitination; however, the exact role for ubiquitination in EGFR endocytosis is not fully elucidated [31]. Even though many adapter proteins (Figure 4(1)), like epidermal growth factor receptor substrate 15 (Eps15, found in clathrin-coated pits), have been shown to be important for EGFR endocytosis [32], the molecular mechanisms behind them wait to become fully enlightened [16,22].

Unoccupied receptors are then quickly recycled back to the membrane, while ligand-bound receptors recycle slowly or degrades only in lysosomes. Sorting into Rab4,35- or Rab8,11-positive early endosome domains promotes fast or slow recycling (cf. section above) accompanied by continuous deubiquitination (Figure 4(2)). On the other side, if ubiquitinated EGFR is recognized by Rabex-5, hepatocyte growth factor-regulated tyrosine kinase substrate (Hrs) and tumor susceptibility gene 101 (Tsg101), EGFR degradation occurs (Figure 4(3)). Hrs and signal transducing adapter molecule (STAM) cling EGFR in early endosomes; subsequently, the whole ESCRT 0-III complexes assemble. Deubiquitinating enzymes (DUBs) remove Ub before EGFR sorting is completed to replenish the free Ub pool. Remarkably, apart from inactivating EGFR, protein tyrosine phosphatase PTP1B co-localizes with the receptor during the endocytic traffic, herby PTP1B dephosphorylates ESCRT members to foster lysosomal degradation [6,16]. Recently, autophagy has been shown to regulate EGFR trafficking from early endosomes and Rab11-mediated recycling [33].

Although initiation of the receptor endocytosis requires the cytoplasmic tail to be present, the process of endosomal retention and progression is independent from intrinsic receptor kinase activity. While in steady state 2–3% of all empty EGFR are internalized per minute, ligand binding accelerated internalization rates and retention of EGF/EGFR within the cell [34]. Internalized EGF/EGFR proceeds towards lysosomal degradation by maturation of early endosomes. While Rab5 is a component of early endosomes, Rab7 defines the functional identity of late endosomes and makes the compartment competent for fusion with the lysosome (degradation competent) [35]. Accordingly, maturation of early endosomes not only includes increasing acidification but also the so-called Rab5–Rab7 conversion. About 35% of Rab5-positive endosomes can be categorized as rapidly maturing, as they accumulate Rab7 with a time constant of 30 s, while the remaining 65% mature much slower, not acquiring Rab7 even after 100 s [21]. It is likely that the increase in Rab7 content arises from fusion with preexisting vesicles. EGF is preferentially targeted directly to the dynamic, rapidly maturing population of early endosomes (86%) with only a small fraction (14%) joining the static, slowly maturing endosomes. In contrast, the iron transporter transferrin undergoes none-selective sorting by early endosomes. Since the static, slowly maturing endosomes constitute a larger population than the dynamic, rapidly maturing ones, most of the transferrin molecules are delivered to the static population [21].

Clinically relevant are EGFR mutants displaying trafficking defects, thereby gaining oncogenic functions. Prominent examples comprise the variants EGFRvIV, EGFRvV, EGFRvIII, and v-ErbB, which are supposed to be deficient in endocytic downregulation [22].

The binding of a ligand to EGF receptors triggers internalization and recycling of both the receptor and portions of the surrounding membrane. This internalization serves as a regulatory mechanism to control signaling by reducing the number of available receptors on the cell surface, particularly impacting the EGFR’s ability to respond to its ligand EGF. Once internalized, the receptor and ligand are sorted within early endosomes, which may either recycle the receptor back to the membrane or direct it for degradation, depending on ubiquitination and other factors.

## 3. Redox Control of the Endosomal Compartment in EGF Signaling

At least a portion of endosomes baring clathrin coats can be sub-grouped into receptosomes (Figure 1), which enable further signaling of the appropriate receptors upon ligand binding [36]. Formation of receptosomes selectively avoids fusion with lysosomes and thereby further degradation of receptors and ligands. An experiment published in 1996 may shed light on a possible redox-mediated sorting procedure of EGF/EGFR complexes. For internalization, uncoated endocytic vesicles fuse with early endosomes as the first line of internalization. After emergence from these vacuoles, EGF/EGFR complexes are delivered exclusively to a preexisting subset of endocytic vacuoles, identified by fluid-phase horseradish peroxidases (HRP) internalized several hours before. Delivery of EGF to these preloaded vacuoles was inhibited by incubating living cells with DAB and H_2_O_2_. Consequently, EGF degradation was inhibited and late endosomes accumulated, while internalization of EGF proceeded with unaltered kinetics. In contrast, both, the internalization and recycling of transferrin, which served as a control, were unaffected [37]. Although we do not know what concentration was used in that study, it cannot be excluded that H_2_O_2_ had an effect on endosomal fusion. Lysosomes are characterized by markers such as lysosomal-associated membrane protein 1 (LAMP1) and mannose-6-phosphate receptor [13]. Interestingly, in the paper mentioned above, HRP-loaded vesicles are only positive for LAMP1, indicating that vesicles arising from endocytosis do not express but rather fuse to preformed vesicles containing mannose-6-phosphate receptor. The authors compared EGF- and transferrin-internalization in HEp-2 cells. While EGF-loaded late endosomes fused with HRP-preloaded lysosomes, transferrin-loaded vesicles did not [37]. Together, these data underscore the possibility that EGF/EGFR-loaded late endosomes are distinct from transferrin-loaded late endosomes. Transferrin-loaded late endosomes may mature into lysosomes without fusion or fuse only to lysosomes positive for mannose-6-phosphate receptor, while EGF/EGFR-loaded late endosomes fuse with pre-existing lysosomes negative for mannose-6-phosphate receptor that arise from earlier endocytosis. In the context of redox signaling, it is important to note that a FRET probe-based analysis of the human transferrin receptor 1-mediated endocytosis attested to a reducing environment in the endosomal compartments and the dynamics of transferrin receptor 1 trafficking [38].

In contrast, a group of specialized components of the endosomal compartment, termed “redoxosomes”, are characterized by redox-active signaling and contain internalized, ligand-bound receptors that need reactive oxygen species (ROS) for signal transduction (Figure 5) [39]. The most prominent effectors of ROS signaling are cysteines and disulfide bridges within proteins [40]. It is likely that lysosomes store hydrolases, potentially in an inactive form. Once a substrate has entered a lysosome, the proteolytic steps are probably performed by cysteine endoproteinases [41]. Reduction of disulfide bonds was discovered as a key step in increasing the access of proteolytic processing enzymes to target proteins in lysosomes. Accordingly, lysosomes were thought to be the vesicular compartment that mediates protein disulfide reduction, while it was clear by 1991 that early recycling endosomes do not comprise reducing activity [42]. Later in 2005, Austin et al. found that recycling endosomes, late endosomes, and lysosomes are not reducing but oxidizing and comparable with conditions in the endoplasmic reticulum [43]. These data indicate the existence of two distinct pathways within the endosomal compartment. One of which might be redox-independent, while the other one depends on intra-endosomal ROS formation.

A subset of clathrin-coated endosomes, called receptosomes, enable continued signaling by preventing fusion with lysosomes, thereby avoiding receptor and ligand degradation. Studies show that EGF/EGFR complexes are delivered to specific endocytic vacuoles, distinct from transferrin-containing vesicles, with redox conditions potentially playing a role in their sorting. Furthermore, redoxosomes, a group of specialized endosomal compartments, rely on ROS for signal transduction, suggesting the existence of two distinct endosomal pathways: one that is redox-independent and another dependent on intra-endosomal ROS activity.

### 3.1. Redoxosomes

By nature, endosomes contain at least a portion of the cell membrane and its ROS-forming enzymes (Figure 5), internalized together with the receptor [44]. The major source of membrane-associated ROS is the family of NADPH oxidases, which at present includes seven members (Nox1-5 as well as DUOX1 and 2) [45]. Especially Nox1 and Nox2 together with their associated subunits, p22^phox^ for both, NoxO1 and NoxA1 for Nox1, and p47^phox^ and p67^phox^ for Nox2, have been shown to act on redoxosomes [46,47]. Both Nox1 and Nox2 produce •O_2_ˉ [48]. Nox2, the phagocytic NADPH oxidase is usually recognized in endosomes and lysosomes within the process of unspecific host defense and antigen processing [49,50]. However, in SKOV3 cells treated with lysophosphatidic acid (LPA), internalization of the LPA receptor goes along with recruitment of Nox2 into early endosomes, and subsequent LPA-mediated signaling is abrogated when Nox2 activity is inhibited [51]. Further, Nox2-derived ROS were described to promote the formation of an active interleukin-1/MyD88 complex in the endosomes of MCF-7 cells [52]. Obviously, endosomes containing Nox2 are composed of an internalized plasma membrane and generate •O_2_ˉ into the endosomal lumen to initiate signaling at intracellular sites [53]. A potential mechanism of how the accompanying shift in endosomal charge is prevented was shown in human and murine aortic SMCs. Stimulation of those cells with TNFα and interleukin-1 beta resulted in Nox1 and p22^phox^ dependent ROS production within intracellular vesicles. ClC3 co-localizes with Nox1 in early endosomes and is required for charge neutralization of the electron flow generated by Nox1 across the membrane of signaling endosomes by transporting Clˉ and H^+^ (Figure 5(2)). The importance of this mechanism is certified by the fact that cytokine activation of nuclear factor kappa B in SMCs requires both Nox1 and ClC3 [54].

As pointed out above, H_2_O_2_ is the molecule needed to enable endosomal signaling. Accordingly, endosomes require spontaneous or enzyme-catalyzed dismutation of •O_2_ˉ into H_2_O_2_. Indeed, in addition to intra-endosomal localization of cytosolic SOD1, extracellular superoxide dismutase (ecSOD/SOD3) is internalized in BSA-stimulated murine endothelial cells and co-localizes with early endosome antigen EEA1 [55]. Interestingly, pH changes in the course of endosome progression may provide a feedback mechanism for SOD activity. Reduction of pH to or below 5 in late endosomes forces evasion of SOD1 from endosomes in human endothelial cells [56]. Although it is likely that changes in pH also affect endosomal localization of NADPH oxidases, it remains elusive whether or not they determine the processing of early endosomes. NADPH oxidase Nox4 directly produces H_2_O_2_, and therefore Nox4-mediated signaling is independent from SOD. As Nox4 is mainly expressed in the ER [57], it is likely that Nox4 stems from intracellular structures fused with early endosomes and accordingly may play a role in late endosomal signaling. In contrast, activation of cluster of differentiation (CD)38 in early endosomes appears to be Nox4-dependent [58]. More research is needed to define the role of Nox4 in endosomal transition and signaling.

Endosomes contain parts of the cell membrane and ROS-forming enzymes, with NADPH oxidases, particularly Nox1 and Nox2, being key sources of ROS. These enzymes produce •O_2_ˉ in endosomes, influencing intracellular signaling pathways, such as those involving LPA and interleukin-1 in specific cell types. The dismutation of •O_2_ˉ to H_2_O_2_, facilitated by SOD, is essential for endosomal signaling, and changes in endosomal pH may regulate this process, with Nox4 playing a distinct role in late endosomal signaling by directly producing H_2_O_2_ without SOD involvement.

### 3.2. Redox-Dependent EGFR Trans-Activation

In the case of the EGF/EGFR, NADPH oxidases may only play a role in transactivation of the receptor (Figure 5(1)) [59,60]. For its endosomal signaling, quiescin sulfhydryl oxidase 1 (QSOX1) has been identified as a cellular pro-oxidant, which accelerates ubiquitination-mediated degradation of EGFR and its intracellular endosomal trafficking in hepatocellular carcinoma (HCC) cells [61]. The longer version of human QSOX1 protein (hQSOX1a) is a transmembrane protein localized primarily to the Golgi apparatus [62], while QSOX1b lacks the transmembrane helix and therefore is soluble, secreted, and accumulates in extracellular fluids [63]. Importantly, the enzyme produces H_2_O_2_ that can reduce thiols in proteins and forms disulfides. Another reactive oxygen species is the superoxide anion (•O_2_ˉ). It was suggested that •O_2_ˉ and Clˉ are transported through DIDS-sensitive chloride channel(s) out of early endosomes, where •O_2_ˉ is converted into H_2_O_2_ by superoxide dismutase (SOD1). H_2_O_2_ then enables further signaling outside of the endosome [64]. Such mechanisms would define endosomes as internal sources of ROS and suggest •O_2_ˉ-formation to occur within endosomes. Both cellular and endosomal ROS activate c-Src and matrix metalloproteases (MMP) followed by ligand shedding of EGFR ligands. This EGFR *trans*-activation leads as a feed-forward mechanism to further ROS creation by Nox. As a result, specific signals are generated for proliferation, neointima formation, etc. [65,66].

Together, the data indicate a role of H_2_O_2_ at least in early endosome signaling, which is terminated by a pH-dependent feedback mechanism in the course of endosome progression.

In the case of EGF/EGFR signaling, NADPH oxidases may play a role in transactivation, while the enzyme QSOX1 is a key pro-oxidant that accelerates EGFR degradation and endosomal trafficking in certain cells. QSOX1 produces H_2_O_2_, which is crucial for reducing protein thiols and enabling further signaling outside the endosome. ROS like •O_2_ˉ are converted into H_2_O_2_ within endosomes, driving EGFR transactivation and generating signals for processes like cell proliferation, with a pH-dependent feedback mechanism regulating this signaling as endosomes mature.

## 4. Conclusions

Early endosomes are compartments for fine-tuned sorting and diverse fates that a cargo can be dictated to. The discovery of redoxosomes has added another layer of complexity to the concept of early endosomes as active signaling components. Further research is needed, especially when it comes to EGFR signaling via the endosomal compartment, as trafficking-deficient EGFR mutants harbor oncogenic features [67], which underlines the therapeutic potential of intervening with endosomal sorting.

## Figures and Tables

**Figure 1 antioxidants-13-01215-f001:**
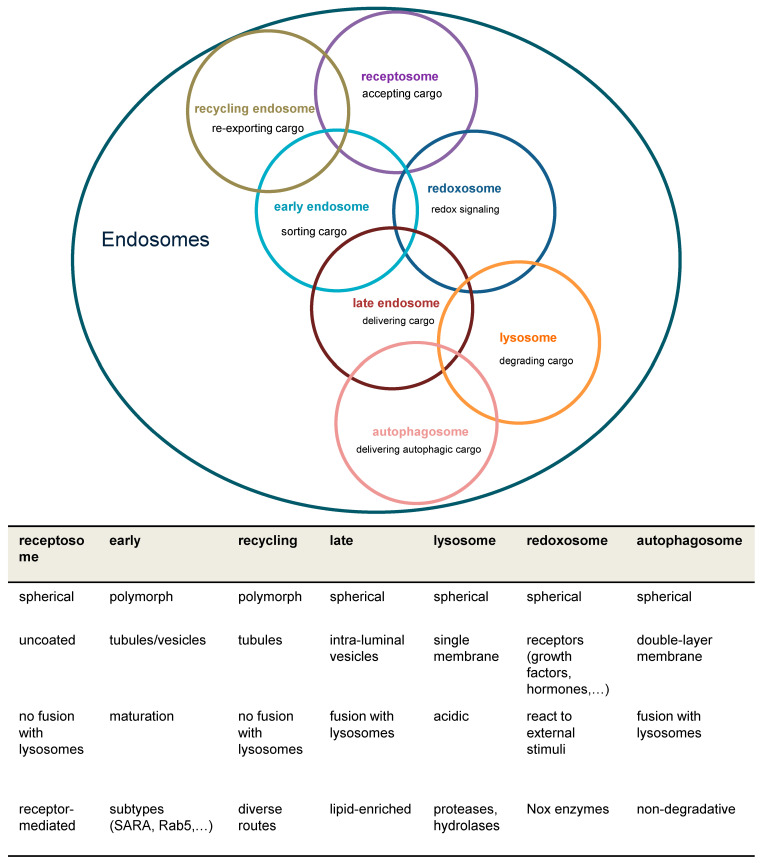
**Overview of major endosomal subtypes and their charactericstic features.** Overlapping circles indicate that individual subspecies of endosomes are not always clearly categorizable. Nox = NADPH oxidase; ROS = reactive oxygen species; SARA = SMAD anchor for receptor activation.

**Figure 2 antioxidants-13-01215-f002:**
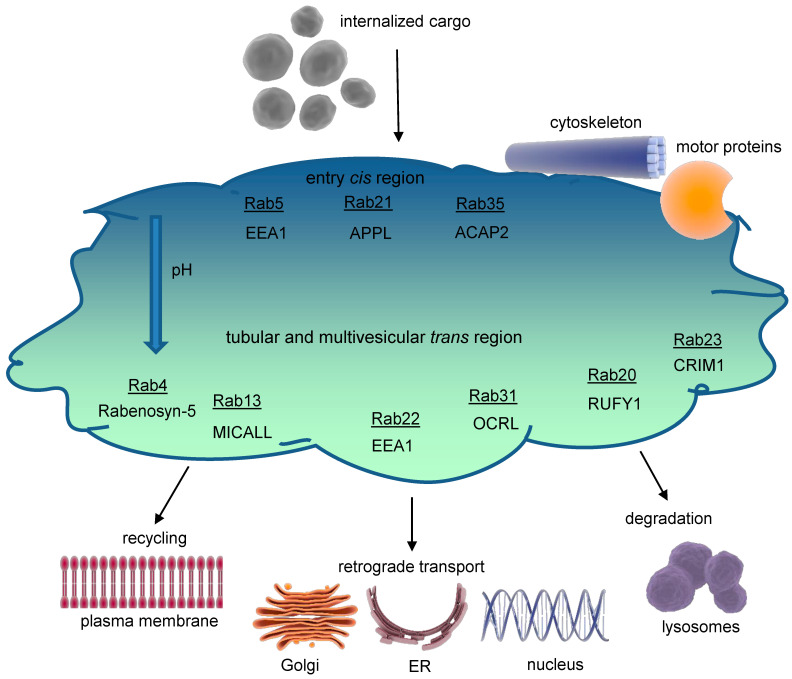
**Structure of the early endosome.** Early endosomes display a dynamic, pleiomorphic appearance with a *cis* site for incoming cargo and a *trans* region with tubular and multivesicular structures that differ in their pH according to the destination of cargo, either for recycling or degradation. The most important Rab proteins and exemplarily one of their effectors are displayed. Cargo exits the early endosome for further intracellular trafficking. Early endosomes are connected to both the microtubule organizing center (MTOC) of the cytoskeleton through annexins and to several motor proteins, which allow movement of early endosomes themselves and cargo delivery. The endosomal pH gradient is indicated by color. ACAP2 = ArfGAP with coiled coils, ankyrin repeat, and PH domains 2; APPL = adaptor protein, phosphotyrosine interacting with PH domain, and leucine zipper 1; EEA1 = early endosomal antigen 1; CRIM1 = cysteine-rich motor neuron 1; MICALL = Mical-like protein; OCRL = inositol polyphosphate 5-phosphatase; Rab = Ras-related protein; RUFY1 = RUN and FYVE domain-containing protein 1.

**Figure 3 antioxidants-13-01215-f003:**
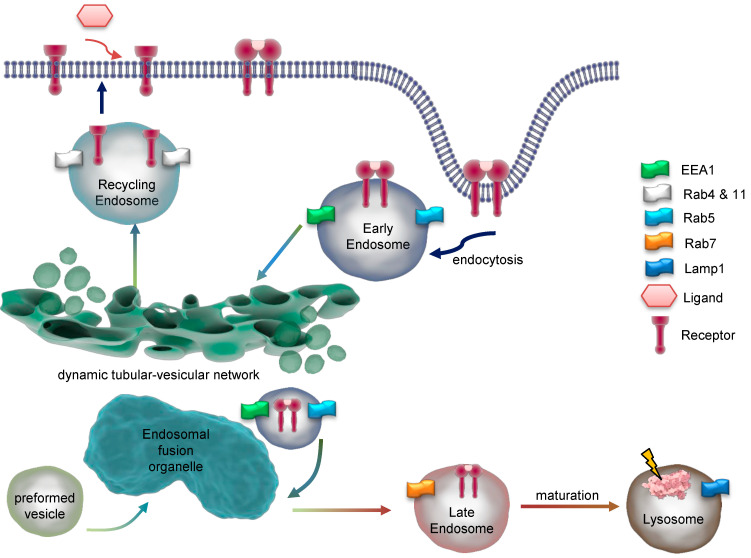
**Dynamics and maturation of endosomes.** Endocytosis of a dimeric receptor after ligand binding is shown. At early endosomes, the receptor is sorted for recycling or degradation and travels through the endosomal network. Key marker proteins of endosomal compartments are indicated. EEA1 = early endosome antigen 1; Rab = Ras-related protein; Lamp1 = lysosomal associated membrane protein 1.

**Figure 4 antioxidants-13-01215-f004:**
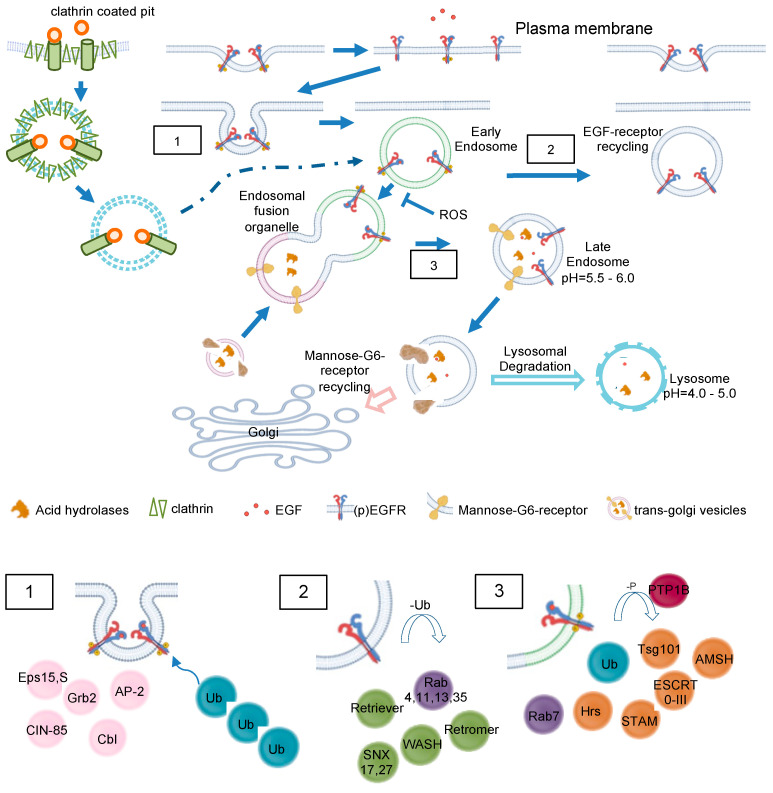
**EGF and MG6 receptor trafficking in detail.** Clathrin-mediated endocytosis of EGFR and intracellular trafficking of M6RR are shown. EGFR may be translocated to other cellular compartments like nucleoplasm or mitochondria (not depicted here). No claim to completeness of molecules is shown. (**1**) Recruitment of major adaptor proteins to EGFR during internalization. The activated receptor dimers are internalized by various pathways depending on ligand type and concentration. The adapter protein Grb2 recruits the ubiquitin ligase Cbl to EGFR, causing mono- or polyubiquitination. Multiple Ub-binding proteins interact with the ubiquitinated EGFR. (**2**) Association of recycling machinery. Association of Rab4,11,13,35 leads to direct or indirect recycling via the endosomal recycling complex. Deubiquitinating enzymes replenish the free Ub pool. (**3**) Lysosomal targeting of EGFR. Dephosphorylation of EGF receptor tyrosine kinases by PTPs terminates signaling and promotes the formation of intraluminal vesicles/multivesicular bodies. EGFR interaction with Hrs and recruitment of the ESCRT complex are necessary for the late endosomal–lysosomal route. AMSH = associated molecule with the SH3 domain of STAM; Cbl = Casitas B-lineage lymphoma protein; CIN-85 = Cbl-interacting protein; Eps= epidermal growth factor receptor substrate 15; ESCRT = endosomal sorting complex required for transport; Hrs = hepatocyte growth factor-regulated tyrosine kinase; PTP1B = Phosphotyrosinephosphatase 1 B; SNX = sorting nexin; STAM = signal transducing adaptor molecule Substrate; Tsg101 = tumor susceptibility gene 101; Ub = ubiquitin; WASH = Wiskott–Aldrich syndrome protein and SCAR homolog.

**Figure 5 antioxidants-13-01215-f005:**
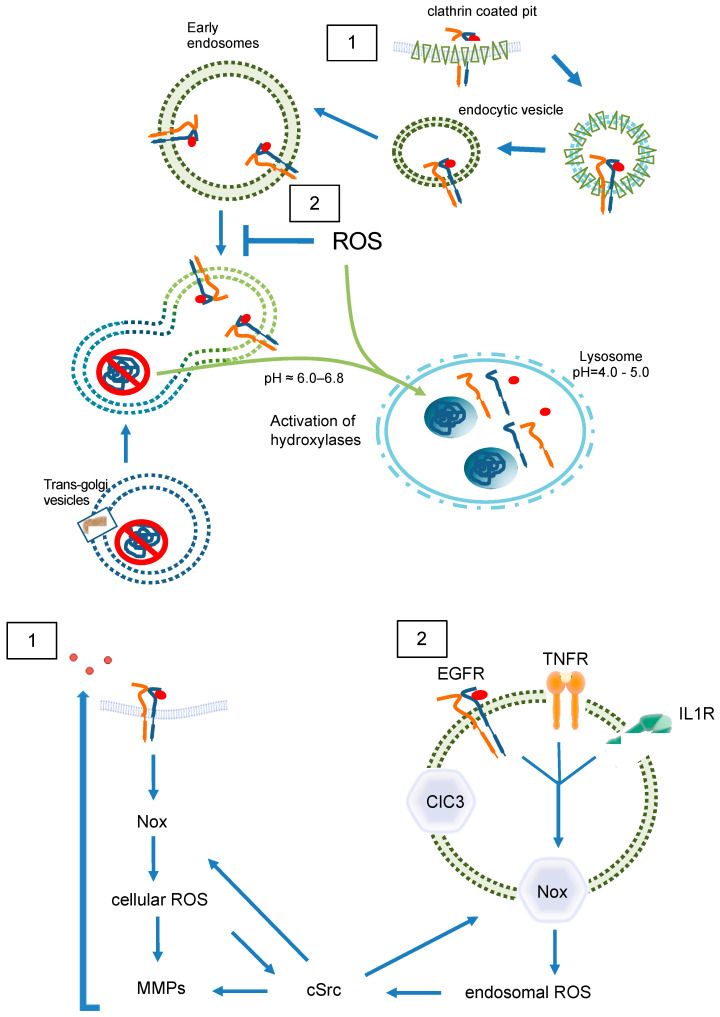
**ROS in endosomal sorting and signaling.** Redox signaling is initiated by EGFR (**1**) at the plasma membrane (**2**) inside of endosomes. Redox-active endosomes (Redoxosomes) harbor Nox enzymes producing ROS downstream of several signaling cascades. Both cellular and endosomal ROS activate c-Src and MMPs followed by shedding of EGFR ligands. This EGFR *trans*-activation leads as a feed forward mechanism to further ROS creation by Nox. The termination of the ROS signaling by dismutation or other means is not part of the figure. ClC3 provides charge neutralization. ClC3 = chloride channel 3; cSrc = cellular sarcoma kinase; EGFR = epidermal growth factor receptor; IL1R = interleukin 1 receptor; MMP = matrix metalloprotease; Nox = NADPH oxidase; ROS = reactive oxygen species; TNFR = tumor necrosis factor receptor.

**Table 1 antioxidants-13-01215-t001:** Classification of receptor mediated signaling.

Type I Signaling:	Type II Signaling	Type III Signaling		
			IIIA	IIIB
Ligand binding to the receptor results in direct signal transduction	Upon binding of the ligand, second messengers mediate signal transduction	Binding of the ligand to its receptor induces internalization and formation of endosomes, which transduced the appropriate signaling cascade	Recruitment of cytosolic proteins to the receptor, followed by internalization and endosome formation	Fusion of the endosome with other intracellular vesicles
e.g., Tyrosinkinase receptors	e.g., Adrenalin β-receptors	e.g., EGF-receptor	e.g., EGF- and TGFβ-receptors	e.g., TNFα-receptor fuses with the Golgi to transduce apoptosis signals

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
