# Peer review of "Redox Signaling in Endosomes Using the Example of EGF Receptors: A Graphical Review"

_antioxidants, 2024, doi:10.3390/antiox13101215_

Round 1

Reviewer 1 Report

The manuscript of Hebschen and Schröder is aimed at summarizing the current knowledge concerning the role of redox modulators in endosomal signaling. To achive their aim, Authors focus their review on one specific receptor, the (epithelial growth factor receptor (EGFR). Authors divide their manuscript into the following 4  main chapters: Introduction, The Endosomal Compartment in EGF Signaling, Redox Control of the Endosomal Compartment in EGF Signaling and Conclusions. Except for the Introduction and Conclusion, chapters are also divided into subchapters providing a well-structured presentation of the literature. Authors complement the manuscript with 5 illustrative Figures and 1 Table and cite 65 references. The Figures are informative and original, and the text is written in a clear, easy-to-follow manner.

The manuscript provides a balanced review on a current and novel topic. The manuscript fits the scope of the “Antioxidants” and is of interest for the readers of the journal. 

This reviewer has only one minor suggestion to improve the manuscript. 

Remarks:

1.    Introduction: 

Authors should cite two recent reviews about Type I and Type II receptor signaling e.g. “Type I and type II receptor signaling have been reviewed in ….”

before the sentence:

In this review, we will concentrate on type III signaling on the example of epithelial growth factor (EGF) and its receptor (epithelial growth factor receptor, EGFR), which involves receptor internalization.”

The manuscript of Hebschen and Schröder is aimed at summarizing the current knowledge concerning the role of redox modulators in endosomal signaling. To achive their aim, Authors focus their review on one specific receptor, the (epithelial growth factor receptor (EGFR). Authors divide their manuscript into the following 4  main chapters: Introduction, The Endosomal Compartment in EGF Signaling, Redox Control of the Endosomal Compartment in EGF Signaling and Conclusions. Except for the Introduction and Conclusion, chapters are also divided into subchapters providing a well-structured presentation of the literature. Authors complement the manuscript with 5 illustrative Figures and 1 Table and cite 65 references. The Figures are informative and original, and the text is written in a clear, easy-to-follow manner.

The manuscript provides a balanced review on a current and novel topic. The manuscript fits the scope of the “Antioxidants” and is of interest for the readers of the journal. 

This reviewer has only one minor suggestion to improve the manuscript. 

Remarks:

1.    Introduction: 

Authors should cite two recent reviews about Type I and Type II receptor signaling e.g. “Type I and type II receptor signaling have been reviewed in ….”

before the sentence:

In this review, we will concentrate on type III signaling on the example of epithelial growth factor (EGF) and its receptor (epithelial growth factor receptor, EGFR), which involves receptor internalization.”

Author Response

Major comments

The manuscript of Hebschen and Schröder is aimed at summarizing the current knowledge concerning the role of redox modulators in endosomal signaling. To achive their aim, Authors focus their review on one specific receptor, the (epithelial growth factor receptor (EGFR). Authors divide their manuscript into the following 4  main chapters: Introduction, The Endosomal Compartment in EGF Signaling, Redox Control of the Endosomal Compartment in EGF Signaling and Conclusions. Except for the Introduction and Conclusion, chapters are also divided into subchapters providing a well-structured presentation of the literature. Authors complement the manuscript with 5 illustrative Figures and 1 Table and cite 65 references. The Figures are informative and original, and the text is written in a clear, easy-to-follow manner.

The manuscript provides a balanced review on a current and novel topic. The manuscript fits the scope of the “Antioxidants” and is of interest for the readers of the journal.

This reviewer has only one minor suggestion to improve the manuscript.

Remarks:

  1. Introduction:

Authors should cite two recent reviews about Type I and Type II receptor signaling e.g. “Type I and type II receptor signaling have been reviewed in ….”

before the sentence:

“In this review, we will concentrate on type III signaling on the example of epithelial growth factor (EGF) and its receptor (epithelial growth factor receptor, EGFR), which involves receptor internalization.”

We thank the reviewer for that valuable advice and referenced two additional reviews:

“Excellent reviews have been published on type I and II signaling (DOI: 10.1136/ard-2023-223850, DOI: 10.1101/cshperspect.a005900).”

Reviewer 2 Report

"This review provides a contemporary overview of the different types and function of endosomes in the cell, focusing in the redox signaling (And EGFR as proof of concept). The authors highlight the different subtypes of endosomes present in the cell, present their structure, dynamics and maturation process. They present very representative infographs of the process that summarizes the process explained.

The review is well-written and employs a relevant and current bibliography. The paper may be of interest to readers in the field of endosomes and scientists working in oxidative stress.

Some of the points discussed in the article delve deeply into the molecular processes taking place within endosomes. However, with the detailed description of various components, the overall flow of the process is sometimes lost. Thus, it would be helpful if each section concluded with a paragraph summarizing the most important concepts.

For this referee, the concept of intracellular movement of endosomes, as discussed in section 1.2 'Structure of Early Endosomes,' is not entirely clear.

In the context of the redox control of the endosomal compartment in EGF signaling, the authors reference an experiment from 1996, stating that 'they do not know the concentration of DAB and H₂O₂ used in the study.' However, several more recent experiments have been published that include the precise concentrations, which could provide the necessary information.

There are also some minor comments about the paper:

-          Page 3 Table 1: In the Table, it should be convenient to present the same type of information at the same level. It would help to present the parameter or type of information present in the table, in the lateral of the table (example; shape, type of couting, composition…).

-          In Figure 1 the circles of the different subtypes of endosomes interact each other. Readers would appreciate an explanation of the types of interactions and relationships between the different endosomes.

-          Page 4 Figure 2. The information about the function of the motor proteins in the endosome is not clear.

-          Page 4 Figure 2. Line 78. “Early” should be “early”.

-          Page 6 Figure 3. Line 153. “endsomal” should be “endosomal”.

-          Page 7 Figure 4. Line 175.  Please, confirm that “MG8” it is OK. Perhaps is 6.

-          Page 7 Figure 4. The symbol employed of clathrin should be included in the legend.

-          Page 7 Figure 4. The employment of the title “EGF receptor trafficking in detail”, after the explanation “No claim to completeness of molecules showed” seems to be contradictory.

-          Page 8 Figure 4. Line 199. The “,” after vesicles, should be a “.·

-          Page 8. Line 213. TGFα should be TGFβ.

-          Page 8.Line 214. The abbreviation of “EGF receptor” (EGFR) have been previously described. It should be employed.

-          Page 11.Line 332. The abbreviation of “TNFα” have been previously described. It should be employed.

-          Page 12.Line 363. The abbreviation of “hydrogen peroxide” (H2O2) have been previously described. It should be employed.

Author Response

Major comments

"This review provides a contemporary overview of the different types and function of endosomes in the cell, focusing in the redox signaling (And EGFR as proof of concept). The authors highlight the different subtypes of endosomes present in the cell, present their structure, dynamics and maturation process. They present very representative infographs of the process that summarizes the process explained.

The review is well-written and employs a relevant and current bibliography. The paper may be of interest to readers in the field of endosomes and scientists working in oxidative stress.

Detail comments

Some of the points discussed in the article delve deeply into the molecular processes taking place within endosomes. However, with the detailed description of various components, the overall flow of the process is sometimes lost. Thus, it would be helpful if each section concluded with a paragraph summarizing the most important concepts.

We thank the reviewer for this suggestion and included a summary at the end of each paragraph.

For this referee, the concept of intracellular movement of endosomes, as discussed in section 1.2 'Structure of Early Endosomes,' is not entirely clear.

In the context of the redox control of the endosomal compartment in EGF signaling, the authors reference an experiment from 1996, stating that 'they do not know the concentration of DAB and H₂O₂ used in the study.' However, several more recent experiments have been published that include the precise concentrations, which could provide the necessary information.

Indeed, many studies have been published after this one. However, the point to be made here was not to figure out the exact concentrations or other things. It is just to say that a role of H2O2 cannot be excluded, even in those early experiments. This appeared important to us because any positive roles of ROS where quite understudied at that time.

There are also some minor comments about the paper:

-          Page 3 Table 1: In the Table, it should be convenient to present the same type of information at the same level. It would help to present the parameter or type of information present in the table, in the lateral of the table (example; shape, type of couting, composition…).

We transposed the table as requested.

-          In Figure 1 the circles of the different subtypes of endosomes interact each other. Readers would appreciate an explanation of the types of interactions and relationships between the different endosomes.

We included an explanation for the overlapping circles.

-          Page 4 Figure 2. The information about the function of the motor proteins in the endosome is not clear.

-          Page 4 Figure 2. Line 78. “Early” should be “early”.

-          Page 6 Figure 3. Line 153. “endsomal” should be “endosomal”.

-          Page 7 Figure 4. Line 175.  Please, confirm that “MG8” it is OK. Perhaps is 6.

-          Page 7 Figure 4. The symbol employed of clathrin should be included in the legend.

-          Page 7 Figure 4. The employment of the title “EGF receptor trafficking in detail”, after the explanation “No claim to completeness of molecules showed” seems to be contradictory.

-          Page 8.Line 214. The abbreviation of “EGF receptor” (EGFR) have been previously described. It should be employed.

-          Page 11.Line 332. The abbreviation of “TNFα” have been previously described. It should be employed.

-          Page 12.Line 363. The abbreviation of “hydrogen peroxide” (H2O2) have been previously described. It should be employed.

All done. We thank the reviewer for careful reading of the manuscript.

-          Page 8 Figure 4. Line 199. The “,” after vesicles, should be a “.·

We disagree because the part after the comma relates to the part before the comma and both belong to the same sentence.

-          Page 8. Line 213. TGFα should be TGFβ.

We think it should be TGFa (àhttps://doi.org/10.1038/nrneph.2016.91) and would like to leave this as it is.